# The Prevalence of Diabetes among Hypertensive Polish in Relation to Sex-Difference in Body Mass Index, Waist Circumference, Body Fat Percentage and Age

**DOI:** 10.3390/ijerph19159458

**Published:** 2022-08-02

**Authors:** Anna Maria Bednarek, Aleksander Jerzy Owczarek, Anna Chudek, Agnieszka Almgren-Rachtan, Katarzyna Wieczorowska-Tobis, Magdalena Olszanecka-Glinianowicz, Jerzy Chudek

**Affiliations:** 1First Department of Cardiology, Medical University of Silesia in Katowice, 40-635 Katowice, Poland; 2Health Promotion and Obesity Management Unit, Department of Pathophysiology, Faculty of Medical Sciences in Katowice, Medical University of Silesia in Katowice, 40-752 Katowice, Poland; aowczarek@sum.edu.pl (A.J.O.); magolsza@gmail.com (M.O.-G.); 3Department of Pharmacovigilance, Europharma Rachtan Co., Ltd., 40-061 Katowice, Poland; anna.m.chudek@gmail.com (A.C.); agnieszka@europharma.edu.pl (A.A.-R.); 4Laboratory for Geriatric Medicine, Department of Palliative Medicine, University of Medical Sciences, 60-355 Poznan, Poland; kwt@tobis.pl; 5Department of Internal Medicine and Oncological Chemotherapy, Faculty of Medical Sciences in Katowice, Medical University of Silesia in Katowice, 40-029 Katowice, Poland; chj@poczta.fm

**Keywords:** obesity, visceral obesity, epidemiology, sex-specific effect, physical activity

## Abstract

Background: Little is known about sex differences in the risk of type 2 diabetes (DM2) development related to body fat depot. The main aim of this study was to assess sex-specific differences in the prevalence of diabetes in the relation to body mass, body mass index (BMI), waist circumference (WC), and calculated body fat percentage (BF), adjusted by physical activity, in younger and older hypertensive adults. Subjects/Methods: The survey enrolled 12,289 adult hypertensive outpatients with body weight, height, and WC reported by their physicians across Poland. Prevalence of diabetes was plotted against body mass, BMI, WC, and calculated BF and adjusted by the self-reported level of physical activity. Results: In our cohort, younger women (<60 years) with BMI < 25.0 kg/m^2^ had lower adjusted prevalence of diabetes than corresponding men (3.4% vs. 6.5%), while among older (≥60 years) with BMI < 25.0 kg/m^2^, the prevalence of diabetes was greater in women than in men (19.4% vs. 11.2%). A 25% probability of diabetes was observed for younger women with lower BMI than younger men (32.1 kg/m^2^ and 35.3 kg/m^2^, respectively) and WC (100.7 cm and 116.1 cm, respectively) but greater BF (45.5% and 38.9%, respectively). The corresponding differences in BMI and WC in older ones were much smaller (27.6 kg/m^2^ and 27.2 kg/m^2^, respectively; 83.7 cm and 85.6 cm, respectively), but not for BF (40.7% and 30.1%, respectively). A doubling of diabetes probability (from 25% to 50%), adjusted by physical activity, was attributable to the lower increase in BMI and WC and BF in women than in men (6.3 vs. 9.8 kg/m^2^, 25.0 vs. 36.1 cm, and 6.5 vs. 10.8%, for younger, and 8.1 vs. 11.3 kg/m^2^, 26.2 vs. 73.2 cm and 8.8 vs. 13.3%, for older). Conclusions: This study shows a lower probability of diabetes in younger women than younger men with normal weight BMI ranges, adjusted to physical activity. This probability is greater for hypertensive women, regardless of age, due to the increase in BMI/WC and BF values adjusted for physical activity.

## 1. Introduction

Obesity is the sixth risk of death in the world. The World Health Organization (WHO) reports that obesity affects 400 million, and over 1.6 billion adults are overweight. In Poland, 40% of people aged 25 to 29 years, 55% of people aged 35 to 39 years, and more than 65% of people aged older than 55 years suffer from overweight and obesity [1]. Growing body fat depot, especially in the visceral area, is a cause of the increasing prevalence of hypertension, dyslipidemia, insulin resistance, type 2 diabetes mellitus (DM2), and numerous other complications of obesity. An accumulation of visceral adipose tissue is highly correlated with insulin resistance and fatty liver disease as well as fatty muscle, which contributes to the development of DM2, negatively affects the quality of life, and increases mortality [2]. As was reported by the WHO, the number of people with diabetes all over the world increased from 108 million patients in 1980 to 422 million in 2014 (WHO, 2016). About 90% of people with DM2 are overweight or obese following the WHO criteria [3].

The increase in body mass, body mass index (BMI), and waist circumference (WC) over time is a major risk factor for the development of DM2, both in men and women. Aging is strongly associated with weight gain related to the accumulation of body fat depot during a lifetime. The adverse changes in body composition accelerate in women during the menopausal transition [4]. The age-associated decline in physical capacity related among others to the decrease in especially appendicular skeletal muscle mass [5]. As a consequence of diminishing muscle mass, older adults more frequently develop sarcopenic obesity, which becomes twice as frequent in older men than in women, according to the National Health and Nutrition Examination Survey III, based on sex-specific cut-offs [6]. This reflects a greater slope decline of muscle mass in men.

However, little is known about sex differences in the risk of DM2 development related to the increase in body fat depot. One longitudinal study showed a stronger association between weight gain and increase in BMI and WC and the risk of DM2 development in men than in women; however, sex-specific differences had limited significance [7]. Contrarily, a cohort Japanese study suggested that WC increase was associated with a higher risk of DM2 development among women; however, the difference between women and men was not significant [8].

The sex differences in the DM2 development in overweight and obese subjects may result from a different fat depot distribution [9], higher metabolic activity of visceral adipose tissue, estrogen-dependent stimulation of fatty acids oxidation in the liver [10], and physical activity [11]. In the abdominal region, subcutaneous fat dominates over visceral fat in young and middle-aged (up to 60 years) women. This proportion changes in obese women, especially after menopause when estrogen activity decreases. In men, visceral adipose tissue typically dominates. It is well-established that visceral obesity is a stronger risk factor for the development of DM2 than general obesity assessment [12]. Sex differences may be attributable not only to biological differences but also to social and cultural factors [13]. Of note, BMI, as a diagnostic criterion of obesity, often overestimates body fat mass in men compared with women because men generally have greater muscle mass. Insulin resistance has been closely related to fat distribution in the visceral area that is commonly found in men [14]. Thus, WC has been suggested as a better predictor of DM2 development than BMI in both genders [15].

In overweight women of reproductive age, a greater proportion of fatty acid oxidation is estrogen dependent. The use of carbohydrates as a source of energy during exercise can reduce the risk of liver steatosis and the development of non-alcoholic fatty liver disease (NAFLD) [16]. It is well-known that physical inactivity and a sedentary lifestyle increase the risk of NAFLD [17] and, in turn, DM2 [18].

Concerning the above-mentioned differences between the sexes, we have tested a potentially distinct association between body mass, BMI/WC, and body fat depot, adjusted to declared physical activity and the prevalence of diabetes among men and women participating in a large survey (2017–2018) performed by physicians, with the participation of 14,200 unselected hypertensive outpatients [19].

Regarding the aforementioned gender differences, we believe that differences in anthropometric measures of body fat deposition, and physical activity in men and women may be important predictors of diabetes prevalence.

The main aim of this study was to show sex-specific differences in the prevalence of diabetes in the relation to body weight, BMI, WC, and estimated BF, adjusted for physical activity in young and older hypertensive adults.

## 2. Materials and Methods

The survey was performed in 2017 by 570 physicians and medical trainees managing outpatients with hypertension across Poland. The co-investigators (physicians) were recruited via the Internet using e-mails of physicians that participated in previous surveys. The study group included 14,200 hypertensive Caucasian adult (mean age, 63 ± 12 years; range, 20–93 years) coming back to outpatient clinics for routine control visits. The only inclusion criterion was pharmacologically treated hypertension. There was no exclusion criterion except pregnancy. Among study participants, complete anthropometric measurements data sets were available for 12,289. Only these subjects were included in the presented analysis (Figure 1) and stratified according to the presence of diabetes, obesity, visceral obesity, and age (younger < 60 years old and older ≥ 60 years old).

The questionnaire, completed by physicians (study co-investigators) during a single visit, included age, sex, two attended office blood pressure measurements (performed with a validated monitor adjusted to the arm circumference cuff in a seated position, on the right upper arm, after at least 5 min of rest and at 2 min intervals), body mass, height, and WC (with an accuracy of 0.1 kg, 0.5 cm, and 1.0 cm, respectively), and patients’ medical history, such as smoking/alcohol drinking habits, level of physical activity (physically active/sedentary lifestyle), duration and current pharmacotherapy of hypertension, and comorbidities (diabetes; hypercholesterolemia, defined based on total cholesterol level; coronary artery disease; peripheral artery disease; heart failure; severe chronic kidney disease). The survey was performed by physicians in paper form. Body mass, height, and WC were measured by the investigators.

The study was carried out following national and international standards for research (Helsinki Declaration). The study protocol was accepted by the Bioethics Committee of Medical University of Silesia waved obtaining of the informed consent from survey participants (PCB/CBN/0022/KB/246/021). Each study participant gave vocal consent for the participation in the survey. This fact was witnessed by a physician that performed the survey.

### 2.1. Data Analysis

Nutritional status was defined based on BMI, according to the WHO criteria [20]: underweight (<18.5 kg/m^2^), normal weight (18.5–24.9 kg/m^2^), overweight (25.0–29.9 kg/m^2^), and obesity (≥30.0 kg/m^2^).

The WC thresholds for visceral obesity in Caucasians were as defined according to the International Diabetes Federation (IDF) (≥94 cm in men and ≥80 cm in women), and the National Cholesterol Education Program Adult Treatment Panel III—NCEP ATP III (≥102 cm in men and ≥88 cm in women) [21]. According to them, the WC range was divided into three intervals: below 80 cm, from 80 to 87 cm, and ≥88 cm (for women); and below 94 cm, from 94 to 101 cm, and ≥102 cm (for men), corresponding to those the lack of visceral obesity, mild visceral obesity, and severe visceral obesity, respectively.

Body fat (%) was calculated according to Palafolls formula [(BF = (BMI / WC × 10) + BMI + (10 × sex)], where sex = 1 for women and 0 for men [22].

Diagnosis of diabetes (without separation for DM1 and DM2) was based on the reported antidiabetic therapy. As the study was designed to analyze the effectiveness of antihypertensive therapy, no details concerning antidiabetic medication (drugs’ names and doses) were collected. Coronary artery disease (CAD) was defined as a history of acute myocardial infarction, percutaneous coronary intervention, coronary artery bypass graft, or the occurrence of symptoms of angina pectoris. The clinical diagnosis of peripheral artery disease (PAD) was based on patients’ symptoms (intermittent claudication), revascularization procedures, and imaging, if performed and available. The diagnosis of heart failure (HF) was based on clinical symptoms regardless of the ejection fraction (restricted/preserved) of the left ventricle, if measured. Severe chronic kidney disease (CKD) was defined as the occurrence of estimated glomerular filtration rate (eGFR) below 30 mL/min/1.73 m^2^ and/or proteinuria over 300 mg/24 h. Hypercholesterolemia was defined as cholesterol levels of 190 mg/dl or higher or the use of statins.

A sedentary lifestyle was defined as the lack of regular walking and/or practicing sports. The prevalence of diabetes was plotted against body weight, BMI, WC, and estimated BF and adjusted by the self-reported level of physical activity, sex, and age groups.

We calculated values for body weight, BMI, WC and estimated BF corresponding to 25% (moderate risk for the prevalence of diabetes) and 50% (very high risk for the prevalence of diabetes).

### 2.2. Statistical Analysis

Statistical analyses were performed using STATISTICA 13.0 PL (TIBCO Software Inc., Palo Alto, CA, USA) and Stata SE 13.0 (StataCorp LP, College Station, TX, USA). Statistical significance was set at a *p*-value below 0.05. All tests were two-tailed. Imputations were not performed for missing data. Nominal and ordinal data were expressed as percentages. Interval data were expressed as mean value ± standard deviation in the case of normal distribution. In the case of data with skewed or non-normal distribution, they were expressed as the median, with lower and upper quartiles. The distribution of variables was evaluated by the Anderson–Darling test and the quantile–quantile (Q–Q) plot. Homogeneity of variances was assessed by the Levene test. Nominal and ordinal data were compared with the χ^2^ test. Interval data were compared with the Student’s t-test for independent groups. Differences in body mass, BMI/WC, and fat percentage between the two groups were expressed as the mean difference with a 95% confidence interval. Univariate and multivariable logistic-regression analyses were used to assess factors associated with the occurrence of diabetes. All significant univariable factors were included in the multivariable models. Univariable logistic regression was also used to assess the probability of diabetes occurrence depending on body mass, BMI/WC, and fat percentage, and the increase in diabetes probability for each of these variables, as well as the association of different factors with diabetes, was shown as odds ratios (OR) with 95% confidence intervals (CI) and corresponding *p* values. In addition, to calculate the probability (*p_DM_*) of diabetes occurrence, a standard equation was used:
pDM=1(1+e−β·x), 

where *β* stands for regression factor, and *x* is body mass, BMI/WC, or fat percentage. Based on the calculated equations, we set a lower quartile (25%) of probability and median (50%) of probability as cutoff points, to show related values of body mass, BMI/WC, or fat percentage. We did not establish an upper quartile (75%) of the probability of diabetes occurrence, because not all curves reached this probability value. Moreover, taking into account the sex- and age-related differences in physical activity, we have adjusted the analyses for a sedentary lifestyle. The adjustment only marginally affected the results (less than 2%). Therefore, we have omitted the presentation of unadjusted analyses.

## 3. Results

### 3.1. Analysed Group Characteristics

The analysis included 6163 women and 6126 men who used antihypertensive therapy. In this group, the occurrence of diabetes (26.0% vs. 22.4%; *p* < 0.001) and visceral obesity, regardless of the used criteria, IDF (72.8% vs. 59.9%; *p* < 0.001), and NCEP ATP III (53.4% vs. 32.2%; *p* < 0.001), were more frequent among women and corresponded to a greater calculated body fat percentage (42.3 ± 5.5% vs. 33.1 ± 5.0%; *p* < 0.001) and declared sedentary lifestyle (79.6% vs. 72.4%; *p* < 0.001). Notwithstanding, the percentage of patients with obesity, diagnosed based on the BMI criteria, was lower in women (39.9% vs. 45.6%; *p* < 0.001). As could be expected, smoking and frequent alcohol consumption were more frequently declared by men (45.5% vs. 25.4% and 48.1 vs. 17.7%, respectively; *p* < 0.001 for both).

There were small differences in body mass and calculated body fat percentage between age subgroups. Older men in relation to younger ones had lower body mass by 0.9 kg (95% CI: 0.2–1.6), but greater BMI by 0.6 kg/m^2^ (95% CI: 0.4–0.8), by WC 0.6 cm (95% CI: −0.1 to 1.3), and calculated fat percentage by 0.7% (95% CI: 0.4–0.9). This small difference in BMI resulted in a 5.6% shift in the occurrence of obese and only 0.6% of being viscerally obese (IDF). Older women in relation to younger ones had a greater body mass by 1.6 kg (95% CI: 0.9–2.3), BMI by 1.5 kg/m^2^ (95% CI: 1.2–1.8), WC by 3.0 cm (95% CI: 2.3–3.7), and calculated fat percentage by 1.6% (95% CI: 1.3–1.9). These differences were followed by a 12.5% shift in the occurrence of obesity (WHO) and 5.9% of being viscerally obese (IDF). Aging was associated with a sedentary lifestyle, both in men and women (Table 1).

### 3.2. Diabetic Patients

Among 2980 hypertensive patients with diabetes, there were more women (N = 1605; 53.9%) than men (N = 1375; 46.1%). Diabetic patients were older (65 ± 12 vs. 58 ± 13 years), more frequently obese (62.4% vs. 35.2%; *p* < 0.001) and viscerally obese, both according to IDF (79.7% vs. 62.1%) and NCEP ATP III criteria (58.0 vs. 38.0%, and more often declared sedentary lifestyle (83.1% vs. 73.7%) than non-diabetic (Table 2).

The crude occurrence of diabetes was related to the nutritional status (Figure 2) and was greater among both women and men aged 60 years and older than among younger ones (31.7% vs. 17.9% and 28.7% vs. 16.5%, respectively; *p* < 0.05) (Table 1).

### 3.3. Sex-Specific Age-Stratified Differences in the Prevalence of Diabetes in Relation to Anthropometric Parameters

As expected, the older subgroups had a greater risk of diabetes occurrence for similar ranges of body mass, BMI, WC, and calculated fat percentages. As presented in Table 3, a 25% (lower quartile) probability of diabetes was reached for higher body mass, BMI values, WC, and calculated body fat percentage in younger than older, for both women and men (Figure 3, Table 3).

The doubling of the diabetes probability (from 25% to 50%) was attributable to the lower increase in body mass, WC, and BMI in women than in men (15.1 kg and 25.9 kg, 25.0 cm and 26.2 cm, and 6.3 kg/m^2^ and 8.1 kg/m^2^ in younger and older women, respectively, and 37.8 kg and 68.5 kg, 36.1 cm, and 73.2 cm, and 9.8 kg/m^2^ and 11.3 kg/m^2^ in younger and older men, respectively) (Figure 3 and Table 3). Smaller differences, associated with a doubling in the probability of diabetes, between sex and age groups were observed in attributable gains in body fat percentages: 6.5% in younger and 8.8% in older women and 10.8% in younger and 13.3% in older men.

The calculated ORs for 10 kg increase in body mass, 5 kg/m^2^ increase in BMI, 8 cm increase in WC, and 5% increase in fat percentage were greater for women and younger subgroups, in general (Figure 3). Age differences were only significant for body mass in women [OR = 2.06 95%CI: (1.80–2.36) vs. 1.53 (1.37–1.71)].

As shown in Figure 3, the age differences were decreasing along with increasing body mass, BMI, WC, and fat percentage.

### 3.4. Prevalence of Diabetes in Specific BMI and WC Cutoff Points in Men and Women (Adjusted for Lifestyle)

The probability of diabetes with a BMI of 25 kg/m^2^ was lower by 1.2% and 4.0% in younger and older women, respectively, in comparison to men. Contrary, BMI of 30 kg/m^2^ was higher in women than in men (by 2.8% and 3.8%) and the differences were even much greater for a BMI of 35 kg/m^2^ (by 11.6% and 14.5%) (Figure 3).

In the analysis of cut-off values for WC, the probability of diabetes was lower both in younger and older women for IDF (by 2.6% and 5.2%) as well as for NCEP ATP III criteria (by 1.8% and 1.4%) (Figure 3).

### 3.5. Association between Sex and the Prevalence of Diabetes in Logistic-Regression Analyses

Concerning the results of univariable logistic-regression analyses, men had a lower risk of diabetes occurrence. After adjustment for age categories, sedentary lifestyle, and frequent alcohol consumption, the male gender was associated with a lower risk in the model with BMI ≥ 30 kg/m^2^ but higher in the model with visceral obesity, according to NCEP ATP III criteria (Table 4).

## 4. Discussion

Our study demonstrates the existence of sex-specific differences in the development of diabetes mellitus in relation to body fat depot measures, including BMI, WC, and body fat percentage estimates, adjusted to physical activity in a huge cohort of hypertensive subjects.

As shown in our study, the prevalence of diabetes associated with anthropometric parameters has the shape of an S curve. Young hypertensive women with normal weight have a lower prevalence of diabetes in relation to men, in the analysis adjusted for a sedentary lifestyle. However, with increasing values for BMI, WC, and estimated body fat percentage, the prevalence increased more in women than in men.

Contrary, in older normal-weight women, the prevalence of diabetes was greater than in men, after adjustment to a sedentary lifestyle. In addition, with increasing body fat percentage, the prevalence was increasing more in women than in men, like in the younger group.

These results support the existence of protective mechanisms declining the occurrence of diabetes only in young women with mild body fat depot. We showed that both younger and older hypertensive women require a higher percentage of body fat than men to increase their probability of developing diabetes by 25%. Body fat percentage associated with a 25% risk of diabetes was greater by 6.7% in those younger than 60 years and by 10.6% in older women, adjusted for a sedentary lifestyle. However, a further increase in fat depot was associated with a greater increase in the probability of diabetes among women than in men. The further gain of body fat percentages by 6.5% in younger and 8.8% in older women and 10.8% in younger and 13.3% in older men was associated with doubling the risk of diabetes from 25% to 50%. This observation clearly suggests that in hypertensive women the protective effect against the development of diabetes in women cannot be attributed to estrogens exclusively, as it was also observed in 60-year-olds and older (the last menstrual bleeding of women statistically occurs between 50–52 years of age). It should be noted that our previously published study has shown that during a 5-year follow-up, a mean 2 kg weight gain was associated with an 8 cm increase in WC in premenopausal normal-weight women. Thus, during the aging process in women, slight weight gain is accompanied by a significant increase in the visceral fat deposit. Furthermore, this study has shown that increasing visceral fat deposit is accompanied by an exacerbation of systemic microinflammation [23]. It seems that these changes are related to declining physical activity rather than hormonal changes. This hypothesis is also supported by the results of the presented study, which has shown a higher percentage of subjects living a sedentary lifestyle among those with diabetes and among women than men. The significance of low physical activity for the accumulation of fat deposits has been described in numerous studies, the most important of which are those describing metabolic obesity with normal weight (MONW) and sarcopenic obesity, which is characterized by decreased muscle mass that is coupled with high levels of adiposity [24]. The diminishing physical activity is partially related to the termination of professional activity. In Poland, the average retirement age is 57 years for women and 63 years for men [25]. It should be also noted that the percentage of subjects with a sedentary lifestyle in our study hypertensive cohort was higher than in the general Polish population (83.1% among subjects with diabetes and 73.7% without vs. 60% [26]).

The frequency of obesity and diabetes in our study cohort was much higher than in the general Polish population, estimated at 24% (23.4% for women and 24.2% for men vs. 39.9% for women and 45.6% for men) and 6.0% (5.8% for women and 6.2% for men vs. 26.0% for women and 22.4% for men) [26,27]. It is well-known that hypertension is associated with obesity and insulin resistance. Therefore, hypertensives are at greater risk of diabetes development than normotensive patients [28] Moreover, in both the general and our population, the occurrence of diabetes increases with age. This is in accordance with data published in 2017, showing that type 2 diabetes affected 6.38% of the world’s population, 4.4% of those aged 15–49 years, 15% of those aged 50–69, and 22% of those aged 70+ [29]. Quite similar is the prevalence of diabetes in Polish older adults, which was estimated at 23.1% [30]. The factors explaining an increasing occurrence with age are complex and the main ones include increasing incidence of obesity, especially visceral, and muscle loss being partly related to declining physical activity throughout the lifespan [31,32].

Thus, as was expected, obesity diagnosis based on WHO criteria was overrepresented in our cohort of hypertensive patients (in 39.9% of women and 45.6% of men), compared to the general Polish adult population (20–74 years) estimated at, and was increasing with age [26]. The increase started in the age group of 35–44.9 years, and the growth dynamics were noticeably lower in men than in women [26].

The main limitation is that only patients with treated hypertension were included in the study, probably with overrepresentation of hypertensives and diabetics with an unstable course of these diseases, who require more frequent control visits. In addition, there is a lack of data on the type of diabetes, estrogen levels, and menopausal status. However, given that type 1 diabetes occurs in less than 10% of all the people diagnosed with diabetes [33], this might not have a large impact on the obtained results, whereas the strength of the study is the large study group of hypertensives observed in the conditions of everyday clinical practice. Our results are only comparable with Caucasian hypertensive populations with this same feature.

## 5. Conclusions

This study shows a lower probability of diabetes in younger women than men with normal weight BMI ranges, adjusted to physical activity. This probability is greater for hypertensive women, regardless of age, due to the increase in BMI/WC and BF values adjusted for physical activity. These results suggest the existence of protective mechanisms declining the occurrence of diabetes mostly in younger women with mild body fat depot.

## Figures and Tables

**Figure 1 ijerph-19-09458-f001:**
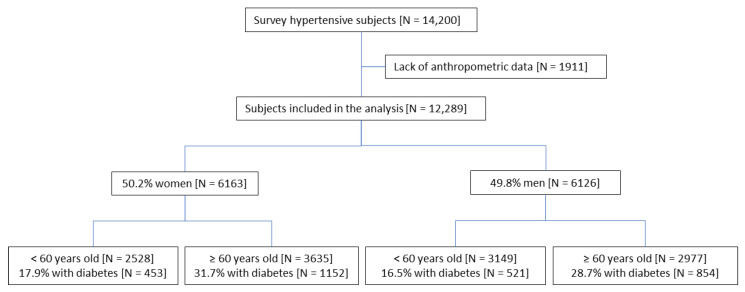
The analysis flow chart.

**Figure 2 ijerph-19-09458-f002:**
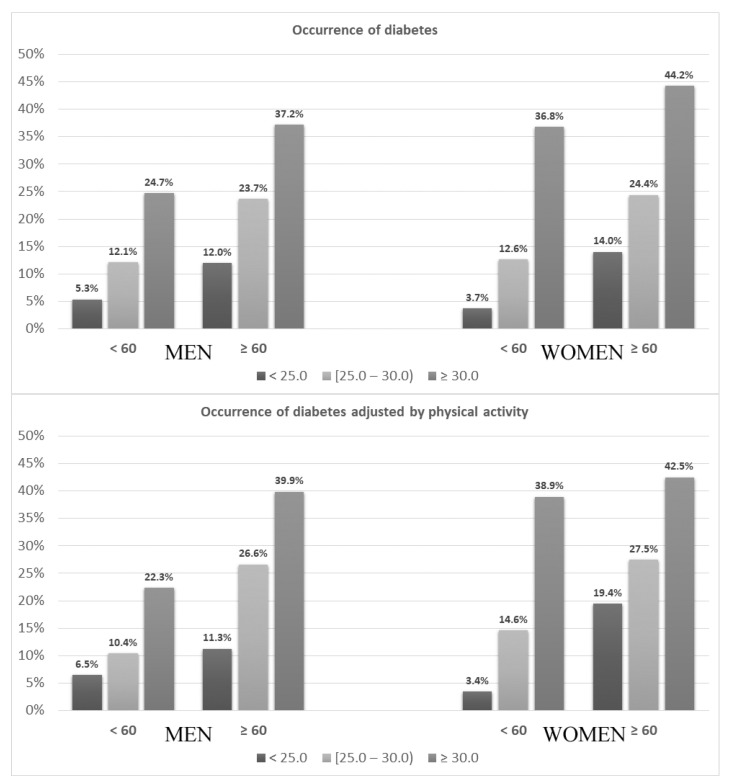
Occurrence of diabetes in younger and older women and men according to BMI categories without (**upper** panel) and after adjustment to physical activity (**lower** panel).

**Figure 3 ijerph-19-09458-f003:**
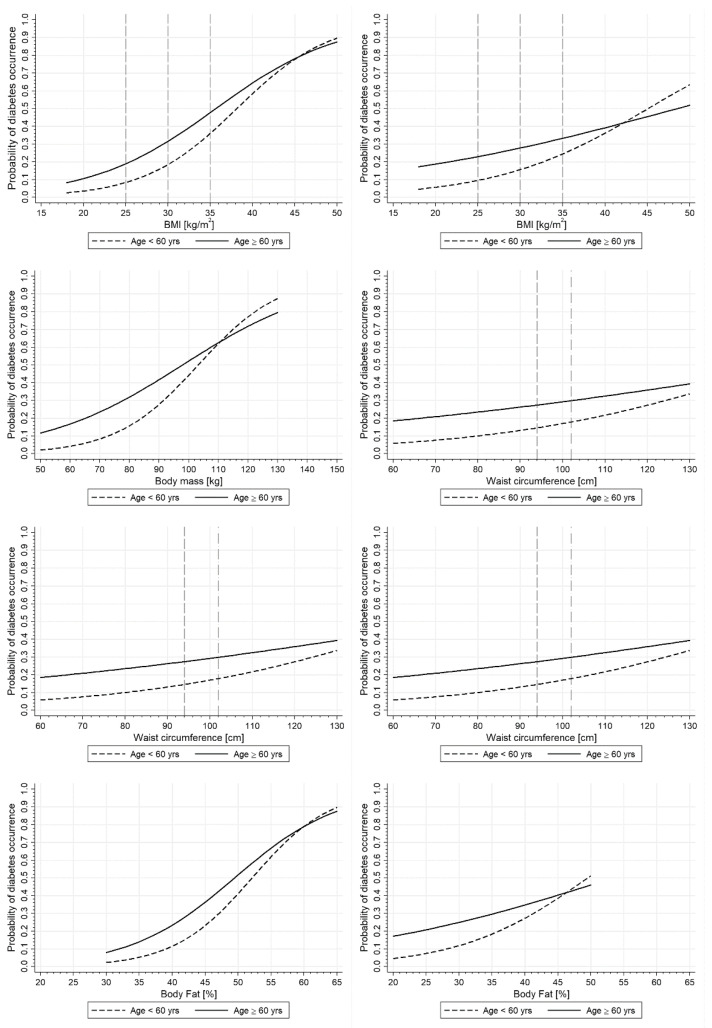
Probability of diabetes occurrence in younger and older men and women in relation to anthropometric parameters. Body mass index (BMI) of 25 kg/m^2^ was associated with the probability of diabetes of 8.3% and 18.9% in younger and older women, respectively, and of 9.5% and 22.9% in younger and older men, respectively. BMI 30 kg/m^2^ with 18.4% and 31.6% in younger and older women, respectively, and 15.6% and 27.8% in younger and older men, respectively. BMI 35 kg/m^2^ with 36.0% and 47.7% in younger and older women, respectively, and 24.4% and 33.2% in younger and older men, respectively. Waist circumference (WC) of 80 cm with 11.9% and 22.2% and 88 cm with 16.0% and 28.5% in younger and older women, respectively. WC of 94 cm with 14.5% and 27.4% and 102 cm with 17.8% and 29.9% in younger and older men, respectively.

**Table 1 ijerph-19-09458-t001:** Characteristics of men and women on antihypertensive pharmacotherapy, stratified by age.

	Men [N = 6126]		Women [N = 6163]	
	<60 Years [N = 3149]	≥60 Years [N = 2977]	<60 Years [N = 2528]	≥60 Years [N = 3635]
Age [years]	48 ± 8	69 ± 7	50 ± 7	70 ± 7
Active or passed smokers [N; %]	1640; 52.1	1148; 38.8 ^#^	813; 32.3	754; 20.9 ^#^
Alcohol consumers [N; %]	1844; 58.9	1103; 37.4 ^#^	586; 23.6	506; 14.1 ^#^
Sedentary lifestyle [N; %]	1715; 67.8	1736; 76.4 ^#^	1510; 76.3	2083; 84.0 ^#^
Body mass [kg]	92.7 ± 13.6	91.8 ± 14.9 *	77.3 ± 13.8	79.0 ± 13.2 ^#^
BMI [kg/m^2^]	29.6 ± 4.4	30.2 ± 4.9 ^#^	28.1 ± 5.3	29.6 ± 5.0 ^#^
Underweight (BMI < 18.5) [N; %]	0	18; 0.6	18; 0.7	0
Normal weight (BMI 18.5–24.9) [N; %]	338; 10.7	292; 9.8	698; 27.6	565; 15.5 ^#^
Overweight (BMI 25.0–29.9) [N; %]	1519; 48.2	1276; 42.9 ^#^	991; 39.2	1435; 39.5
Obesity (BMI ≥ 30) [N; %]	1292; 41.1	1391; 46.7 ^#^	821; 32.5	1635; 45.0 ^#^
Body fat [%]	32.7 ± 4.7	33.4 ± 5.2 ^#^	41.3 ± 5.6	42.9 ± 5.2 ^#^
Waist circumference [WC] [cm]	95.8 ± 13.2	96.4 ± 13.2	87.5 ± 13.4	90.6 ± 14.4 ^#^
WC < 80 (W) and 94 (M) [N; %]	1272; 40.4	1185; 39.8	775; 30.7	902; 24.8 ^#^
WC 80–87.5 (W) and 94–101.5 (M) [N; %]	877; 27.8	819; 27.5	486; 19.2	708; 19.5
WC ≥ 88 (W) and 102 (M) [N; %]	1000; 31.8	973; 32.7	1267; 50.1	2025; 55.7 ^#^
Diabetes [N; %]	521; 16.5	854; 28.7 ^#^	453; 17.9	1152; 31.7 ^#^
Other diseases [N; %]				
Coronary artery disease	628; 19.9	1305; 43.8 ^#^	337; 13.3	1353; 37.2 ^#^
Peripheral artery disease	223; 7.1	388; 13.0 ^#^	69; 2.7	399; 11.0 ^#^
Heart failure	175; 5.6	605; 20.3 ^#^	63; 2.5	539; 14.8 ^#^
Severe chronic kidney disease	142; 4.5	415; 13.9 ^#^	79; 3.1	237; 6.5 ^#^
Hypercholesterolemia	1537; 48.8	1710; 57.4 ^#^	1019; 40.3	1816; 50.0 ^#^

* *p* < 0.01; ^#^ *p* < 0.001.

**Table 2 ijerph-19-09458-t002:** Comparison of patients with and without diabetes.

	Diabetics [N = 2980]	Non-Diabetics [N = 9309]
Age [years]	65 ± 12	58 ± 13 ^#^
Age ≥ 60 years [N; %]	2006; 67.3	4606; 49.5 ^#^
Sex		
Men [N; %]	1375; 46.1	4751; 51.0 ^#^
Women [N; %]	1605; 53.9	4558; 49.0 ^#^
Active or passed smokers [N; %]	1146; 38.7	3209; 34.6 ^#^
Pack-years [N]	24 ± 15	19 ± 13 ^#^
Alcohol consumers [N; %]	795; 27.0	3244; 35.2 ^#^
Sedentary lifestyle [N; %]	2475; 83.1	6862; 73.7 ^#^
Body mass [kg]		
Men	95.0 ± 12.5	91.5 ± 14.6 ^#^
Women	85.1 ± 14.1	75.9 ± 12.4 ^#^
Body mass index (BMI) [kg/m^2^]	31.6 ± 4.7	28.8 ± 4.8 ^#^
Underweight BMI < 18.5 kg/m^2^)	0	36; 0.4 ^#^
Normal weight (BMI 18.5–24.9 kg/m^2^)	158; 5.3	1735; 18.6 ^#^
Overweight (BMI 25.0–29.9 kg/m^2^)	961; 32.2	4260; 45.8 ^#^
Obesity (BMI ≥ 30 kg/m^2^)	1861; 62.4	3278; 35.2 ^#^
Body fat [%]	40.3 ± 7.4	36.8 ± 6.6 ^#^
Waist circumference (WC)		
WC < 80 cm (W) 94 cm (M)	604; 20.3	3530; 37.9 ^#^
WC 80–87.5 cm (W) 94–101.5 cm (M)	647; 21.7	2243; 24.1 ^#^
WC ≥ 88 cm (W) 102 cm (M)	1729; 58.0	3536; 38.0 ^#^
Co-morbidities [N; %]		
Coronary artery disease	1440; 48.3	2183; 23.5 ^#^
Peripheral artery disease	289; 9.7	790; 8.5 *
Heart failure	567; 19.0	815; 8.8 ^#^
Severe chronic kidney disease	377; 12.7	496; 5.3 ^#^
Hypercholesterolemia	1689; 56.7	4393; 47.2 ^#^

* *p* < 0.05; ^#^ *p* < 0.001.

**Table 3 ijerph-19-09458-t003:** The corresponding age- and sex-specific values of anthropometric measurements and estimated body fat for 25% and 50% of the probability of diabetes occurrence (according to logistic-regression models).

	BMI (kg/m^2^)		Body Mass (kg)		Waist Circumference (cm)		Body Fat (%)	
**<60 years**								
**Probability of DM**	**Women**	**Men**	**Women**	**Men**	**Women**	**Men**	**Women**	**Men**
25%	32.1	35.3	88.1	113.5	100.7	116.1	45.6	38.9
50%	38.4	45.1	103.2	151.3	125.7	152.2	52.1	49.7
**≥60 years**								
25%	27.6	27.2	72.0	82.0	83.7	85.5	40.7	30.1
50%	35.7	48.5	97.9	150.5	109.9	158.8	49.5	53.4

**Table 4 ijerph-19-09458-t004:** Results of univariable and multivariable logistic-regression analyses presenting factors associated with the occurrence of diabetes.

	Number (N)	Diabetes N (%)	Univariate OR (95%CI)	Model for BMI ≥ 30 kg/m^2^ OR (95%CI)	Model for WC ≥ 80/94 cm OR (95%CI)	Model for WC ≥ 88/102 cm OR (95%CI)
Age ≥ 60 years	Y (6612)	2006 (30.3)	2.10 (1.93–2.29) ^#^	1.87 (1.70–2.04) ^#^	1.95 (1.79–2.14) ^#^	1.96 (1.79–2.14) ^#^
	N (5677)	974 (17.2)	Ref	Ref	Ref	Ref
Gender	M (N = 6126)	1375 (22.4)	0.82 (0.76–0.89) ^#^	0.89 (0.82–0.98) *	Not significant	1.15 (1.05–1.26) *
	W (N = 6163)	1605 (26.0)	Ref	Ref	Ref	Ref
Alcohol consumers	Y (4039)	795 (19.7)	0.68 (0.62–0.75) ^#^	0.84 (0.76–0.93) ^#^	0.81 (0.73–0.89) ^#^	0.77 (0.70–0.85) ^#^
	N (8250)	2185 (26.5)	Ref	Ref	Ref	Ref
Sedentary lifestyle	Y (9337)	2475 (26.5)	1.75 (1.57–1.94) ^#^	1.44 (1.29–1.61) ^#^	1.50 (1.35–1.68) ^#^	1.50 (1.35–1.68) ^#^
	N (2952)	505 (17.1)	Ref	Ref	Ref	Ref
BMI ≥ 30 kg/m^2^	Y (N = 5139)	1861 (36.2)	6.23 (5.25–7.41) ^#^	2.89 (2.25–3.15) ^#^	–	–
	BMI 18.5–24.9 (1893)	158 (8.3)	Ref	Ref	NI	NI
WC ≥ 80 cm (W) 94 cm (M)	Y (8155)	2376 (29.1)	2.40 (2.18–2.65) ^#^	–	2.29 (2.07–2.53) ^#^	–
	N (4134)	604 (14.6)	Ref	NI	Ref	NI
WC ≥ 88 cm (W) 102 cm (M)	Y (5265)	1729 (32.8)	2.86 (2.58–3.17) ^#^	–	–	2.21 (2.02–2.41) ^#^
	WC < 80/94 (4134)	604 (14.6)	Ref	NI	NI	Ref

* *p* < 0.05; ^#^ *p* < 0.001; NI—not included in the model; Ref—reference group; BMI—body mass index, WC—waist circumference.

## Data Availability

The data presented in this study are available on request from the last author. The data are not publicly available due to privacy restrictions of the funder.

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
