# Peer review of "The Prevalence of Diabetes among Hypertensive Polish in Relation to Sex-Difference in Body Mass Index, Waist Circumference, Body Fat Percentage and Age"

_ijerph, 2022, doi:10.3390/ijerph19159458_

Round 1

Reviewer 1 Report

Even though the idea sounds interesting, there are some important points that need clarification, refinement, reanalysis, rewriting and more information to improve this article. The abstract, introduction, drawing, description of results, and discussion need to be improved to achieve this purpose.

 Major points

1.       The manuscript needs writing and editing. The title should be more concise and concrete, for example, “Sex-specific differences in body mass index, waist circumference, and body fat in Polish patients with diabetes and hypertension”.

The main objective must be direct and the same throughout the manuscript (abstract, introduction, results/discussion). The abstract should be a summary of the main points of this observational study. The first time an abbreviation appears, the full name must be entered, such as body mass index (BMI). Authors should not use the words that appear in the title as keywords. References should be recent, relevant, and referenced correctly.

2.       The introduction section is too concise.  It is not clear what the authors know about this specific topic. It would be better to develop a little more depth on DM in obese/overweight patients according not only to sex but also to age-sex in adults/older adults. Is body fat “BF” right? Please define what "sarcopenic obesity" is. It would be better if the authors offered a clear hypothesis before the main objective of this study. Why would this study be crucial?

3.       The materials and methods section needs significant improvements. What type of study was it? More details are required to be able to replicate this study. The description of the study design should be clear, concise, and detailed. It would be better to show the study design in a flowchart. The authors must declare that the study was carried out following national and international standards for research (Helsinki). The population under study (characteristics) must be defined. It must be clear who obtained the information and by what means (questionnaires), from where the information was obtained (clinical history, face-to-face), whether or not there was a clinical evaluation of the participants, at what time of the year the information was obtained, which were the inclusion (race) and exclusion criteria, and if the population was classified by gender, by age groups, by having diabetes or not, by age group and gender, etc. It would be convenient to show these details in subsections. What was the cut-off point to classify the participants as young and old? What does the classification in three intervals of the WC mean? normal, low risk, high risk? How was activity level measured in daily life? All variables should be defined and measured appropriately.

4.       In the results section: What were the most significant results? Avoid repetition of results (%, CI) in the text if these results are already shown in tables and figures. In Table 1, it would be better to show the results in two different columns (they are underlined in the manuscript). In the legend, it is not necessary to write "$ p < 0.1 and p < 0.01". Table S1 should be displayed in the manuscript rather than in the supplementary material. In Table 2, it would be better to display two different groups (women and men) separately in the same table for better comparison. It would be better to summarize the most significant and relevant results that appear in Table and Figure 2 instead of showing the results that appear in them. Avoid repetition. For whom were the differences much larger for a BMI of 35 kg/m2? What does Ref mean in Table 3? This analysis (lines 258-260) should be described in the materials and methods section. What happened in analysing the other variables studied (CV parameters, cholesterol, GFR, etc.)?

5.       The discussion should be more argumentative. This section should start with the primary objective of this study and the most significant result. This is an observational study. Cause and effect cannot be proven. The authors show the most significant associations between hypertensive patients with diabetes and some anthropometric markers of obesity (body weight, BMI, WC, and BF) and their differences by sex. Pointing out the greater or lesser probability of diabetes appearing in similar populations. The authors did not classify women into these two groups, so how can they support or not support this hypothesis? (Lines 271-273)? Line 277… because it is a specific adult population with hypertension. This particular population suffers from hypertension. Authors must take this detail into account. The characteristics of this disease determine the development of other chronic diseases such as DM2.

Hypertension is twice as frequent in patients with diabetes compared with those who do not have diabetes. Moreover, patients with hypertension often exhibit insulin resistance. They are at greater risk of diabetes development than are normotensive individuals (Petrie, John R et al. “Diabetes, Hypertension, and Cardiovascular Disease: Clinical Insights and Vascular Mechanisms.” The Canadian journal of cardiology vol. 34,5 (2018): 575-584. doi:10.1016/j.cjca.2017.12.005). The results must be discussed from multiple angles and placed in context without being over-interpreted. A paragraph of suggestions should be written before the conclusion. 

6.       The conclusion needs to improve. Review the conclusion based on the results of this study in this specific population.

 I encourage the authors to rewrite the manuscript, thinking about the principal goal of this study, its design and answering with the results and arguments of the discussion the most proper conclusion to this research work.

Reviewer 2 Report

This paper focused on the effect of body mass index (BMI), waist circumference (WC), and body fat on the prevalence of diabetes mellitus among male and female hypertensive patient in Caucasian people. This is a common topic but the authors tried to focus only on hypertensive patient and Caucasian. The result was not presented in clear and detail was too much. Authors need to address several inquiries before reviewer could give a proper decision regarding this paper.

Major revision

1.     In the abstract, authors mentioned that this study used a cross sectional cohort study but in the method section, authors only mentioned survey. Please choose the correct one and do not combine cross sectional with cohort because there is no cross sectional cohort study design.

2.     The aim of this study between abstract and in the main text is not similar and also the title did not explain the article well. Please make the aim similar between abstract and main text and also adjust the title.

3.     In line 88-89, authors stated that they recruited the participants from internet. There is a high possibility to introduce selection bias in this study if authors decide to recruit participants from internet. Please explain how do you address this kind of bias.

4.     This study used a cross sectional study design which is cannot be used to estimate the effect of independent variables to dependent variables because all of the variables were collected in the same time. Please consider to change the word “effect” with “association”.

5.     Author’s main analysis result should be about the association between those three main variables on diabetes mellitus stratified by gender and age. But this kind of result is not currently present in this article, therefore it is not sufficient to answer the objective of this study.

6.     In line 340, authors concluded that there is an association between gender and prevalence of diabetes which is not stated in the main objective of this study. Please consider changing the aim of this study to match the conclusion or change the conclusion to match the aim of this study.

Minor revision

1.     Supplementary table 1 should be in the main article and table 1 or 2 should be in the supplementary because author’s main objective is to estimate the association of BMI, WC, and body fat stratified by gender only.

Round 2

Reviewer 1 Report

Even though this manuscript has improved, there are some important points that the authors must change to publish it. The abstract, introduction, material and methods, description of results, and discussion need to be improved to achieve this purpose.

 Major points

1.     The manuscript needs editing and language correction. The title should be improved, e.g. “The prevalence of diabetes among hypertensive Poles (Polish) in relation to the sex-difference in body mass index, waist circumference, body fat and age”. No need to put "Caucasian". If the authors maintain this characteristic, it is worth mentioning in the discussion section that their results are only comparable with hypertensive populations with this same feature. The authors want to refer to body weight, right? In the conclusion of the abstract: This study shows a lower probability of diabetes in younger women than in men with normal BMI ranges. This probability is greater for hypertensive women, regardless of age, due to the increase in BMI/WC and BF values adjusted for physical activity. It would be better not to show the last sentence here. As the authors have defined the abbreviations for BMI, WC, and BF, it would be best to keep them throughout the manuscript.

2.      The introduction section: It is hard to understand this paragraph: In overweight women of reproductive age, a greater proportion of fatty acid oxidation is estrogen dependent. The use of carbohydrates as a source of energy during exercise can reduce the risk of hepatic steatosis and the development of non-alcoholic fatty liver disease (NAFLD). Right? It would be better if the authors offered a clear hypothesis before the main objective of this study, for example: Regarding the aforementioned gender differences, we believe that differences in body mass, body fat deposition, and physical activity in men and women may be important predictors of diabetes prevalence. Therefore, the main aim of this study was to assess sex-specific differences in the prevalence of diabetes in relation to body weight, body mass index, waist circumference, and estimated body fat percentage, adjusted for physical activity in young and older hypertensive adults.

3.     The materials and methods section needs some improvements. It would be necessary to improve the image of Figure 1 so that it can be seen clearly. Only the first time an abbreviation appears, the full name must be entered (BP). The paragraph on lines 225-237 is not necessary because the authors stated that this study followed national standards for research. It would be better to add this sentence “Prevalence of diabetes was plotted against body weight, BMI, WC, estimated BF and adjusted by the self-reported level of physical activity, sex, and age group.” …in line 266. Wouldn't it be convenient to point to this 25% and 50% of probability as low risk and a higher risk for the prevalence of diabetes?

4.      In the results section: The authors must improve the figures and tables to clearly see their results. Legends should show the meaning of all abbreviations used. Is the difference between 1.2% and 4% (line 538) significant? What does Ref mean in statistical terms?

5.      The discussion should be improved. It would be a good idea not to show this paragraph (lines 582-584) because, with respect to their study design, the authors did not divide the population of women into menopausal or non-menopausal groups. They do not know what percentage of menopausal women is present in the group under 60 years of age. There is no point in comparison. Lines 585-592): It would be a good idea to display your results this way “As shown in our study, the prevalence of diabetes associated with anthropometric parameters has the shape of an S curve. Young hypertensive women with normal weight have a lower prevalence of diabetes in relation to men in the analysis adjusted for a sedentary lifestyle. However, with increasing values for body mass index, waist circumference, and estimated body fat percentage, the prevalence increased more in women than in men. Contrary, in older normal-weight women the prevalence of diabetes was greater than in men, after adjustment to a sedentary lifestyle. In addition, with increasing body fat percentage the prevalence was increasing more in women than in men, like in the younger group.” Lines 595-596: Younger and older hypertensive women require a higher percentage of body fat than men to increase their probability of developing diabetes by 25%. Right? Line 602 “...in hypertensive populations …” Line 621: …premenopausal normal-weight women… Did this group of women have hypertension? Authors must be clear about the population group with which they compare their results. They need to specify this feature. Line 631: decreased muscle mass is coupled with high levels of adiposity (Janice L. Atkins. Chapter 7 - Stéphane Walrand. Effects of Sarcopenic Obesity on Cardiovascular Disease and All-Cause Mortality,Nutrition and Skeletal Muscle, Academic Press, 2019, 93-103). The authors need references to support their results. Please check the discussion text. Line 660: “…might not have…”. Because it is a specific adult population with hypertension. Authors must take this detail into account in their argumentative discussion.

6.      The conclusion needs to improve. Do the authors refer to normal ranges for weight and BMI or just BMI ranges? Make the change in the conclusion of the abstract if necessary.

Minor points are highlighted in the accompanying manuscript.

 I encourage authors to rewrite the manuscript, thinking of specific changes to improve the quality of their manuscript.

Reviewer 2 Report

The article is better now and acceptable to be published in the journal. Authors already revised well the article.

Author Response

Thank you for the revision.